# Enhancing YOLOv5 for Autonomous Driving: Efficient Attention-Based Object Detection on Edge Devices

**DOI:** 10.3390/jimaging11080263

**Published:** 2025-08-08

**Authors:** Mortda A. A. Adam, Jules R. Tapamo

**Affiliations:** School of Engineering, Howard College Campus, University of KwaZulu-Natal, Durban 4041, South Africa; tapamoj@ukzn.ac.za

**Keywords:** autonomous driving, vehicle detection, attention mechanism, lightweight model, object detection, edge devices

## Abstract

On-road vision-based systems rely on object detection to ensure vehicle safety and efficiency, making it an essential component of autonomous driving. Deep learning methods show high performance; however, they often require special hardware due to their large sizes and computational complexity, which makes real-time deployment on edge devices expensive. This study proposes lightweight object detection models based on the YOLOv5s architecture, known for its speed and accuracy. The models integrate advanced channel attention strategies, specifically the ECA module and SE attention blocks, to enhance feature selection while minimizing computational overhead. Four models were developed and trained on the KITTI dataset. The models were analyzed using key evaluation metrics to assess their effectiveness in real-time autonomous driving scenarios, including precision, recall, and mean average precision (mAP). BaseECAx2 emerged as the most efficient model for edge devices, achieving the lowest GFLOPs (13) and smallest model size (9.1 MB) without sacrificing performance. The BaseSE-ECA model demonstrated outstanding accuracy in vehicle detection, reaching a precision of 96.69% and an mAP of 98.4%, making it ideal for high-precision autonomous driving scenarios. We also assessed the models’ robustness in more challenging environments by training and testing them on the BDD-100K dataset. While the models exhibited reduced performance in complex scenarios involving low-light conditions and motion blur, this evaluation highlights potential areas for improvement in challenging real-world driving conditions. This study bridges the gap between affordability and performance, presenting lightweight, cost-effective solutions for integration into real-time autonomous vehicle systems.

## 1. Introduction

Autonomous driving has revolutionized modern transportation by improving road safety, enhancing efficiency, and enabling intelligent real-time decision-making capabilities [1]. Object detection is pivotal in autonomous vehicle perception, enabling vehicles to identify and track surrounding objects accurately. Deep learning-based models, particularly YOLO versions, have emerged as high-performing solutions for real-time object detection as a result of their effectiveness in achieving an optimal trade-off between inference speed and detection accuracy [2,3,4,5]. However, despite their effectiveness, high computational demands, large model sizes, and inference latency make many models unsuitable for resource-constrained edge devices used in autonomous vehicles [6,7,8]. Earlier detection frameworks, such as Faster R-CNN [9], SSD [10], and initial YOLO versions [11], demonstrated promising accuracy but suffered from computational inefficiency that hindered real-time deployment [12,13,14]. More recent architectures, including YOLOv4-5D, YOLOX, and EfficientDet, have sought to improve detection accuracy and efficiency. However, their parameter count and high FLOP requirements continue to pose challenges for deployment in embedded autonomous systems [15,16,17]. Given these limitations, research has increasingly focused on lightweight architectures and attention mechanisms to enhance computational efficiency and detection performance [18,19]. This study introduces enhanced lightweight object detection models derived from YOLOv5s, incorporating Squeeze-and-Excitation (SE) and Efficient Channel Attention (ECA) mechanisms to improve feature selection and computational efficiency. These attention mechanisms improve feature selection while reducing computational overhead and enhancing detection performance in real-time settings. SE modules adaptively recalibrate channel-wise feature responses, strengthening relevant features while suppressing less useful ones [13], while ECA refines attention mechanisms by reducing the complexity of channel dependencies [12]. Training and testing of the proposed models were carried out using the KITTI dataset, a well-established standard for evaluating autonomous driving perception systems. KITTI was chosen due to its well-structured labeling system and diverse yet controlled driving scenarios, making it ideal for assessing model performance in standard conditions. While KITTI provides valuable insights into performance in typical urban and highway environments, it might not effectively encompass the complexities of real-world autonomous driving conditions. To further evaluate model robustness, we evaluated our models on the BDD-100K dataset, which features challenging scenarios, including low-light environments, occlusions, and motion blur. This additional evaluation highlights the model’s strengths and constraints in complex real-life situations. The observed performance gap on BDD-100K is attributed to ECA’s focus on lightweight channel recalibration, which may limit its effectiveness in scenarios requiring more complex spatial modeling. This evaluation underscores the need for future improvements to address challenging environmental conditions in autonomous driving.

The primary contributions of this study are as follows:Proposition of an optimized lightweight object detection model that balances accuracy on edge devices, computational efficiency, and deployment feasibility.Improvement of YOLOv5 by integrating SE and ECA mechanisms into its architecture to enhance feature selection and detection precision without increasing computational overhead.Introduction of extensive performance evaluation on autonomous driving datasets, including the KITTI dataset for standard conditions and the BDD-100K dataset to assess model robustness in challenging scenarios.Benchmark analysis with cutting-edge approaches to lightweight object detection models demonstrates improved trade-offs between computational efficiency and performance.

This study provides a cost-effective yet high-performance solution for edge deployment in autonomous vehicles. Our findings provide an affordable solution with state-of-the-art detection accuracy, enabling more practical real-time deployment in self-driving systems.

Application Scenarios: The proposed models are suitable for edge devices with limited computational capacity. Application areas include autonomous driving perception systems, smart surveillance infrastructures, and IoT-enabled monitoring environments, where balancing computational efficiency, low latency, and detection precision is critical. Our models are particularly suited for embedded AI systems operating in dynamic and resource-limited environments by reducing computational load without compromising detection accuracy. The remainder of this paper is organized as follows: Section 2 reviews related work on lightweight object detection and attention mechanisms. Section 3 describes the proposed methodology and model architecture. Section 4 presents the experimental results along with a comparative performance analysis. Section 5 discusses the implications of the findings, and Section 6 concludes the study while highlighting potential directions for future research.

## 2. Related Work

Deep learning approaches to object detection have achieved impressive results across various applications, making them highly applicable in complex scenarios such as autonomous driving. However, models that achieve high accuracy often incur substantial computational costs. To mitigate this, researchers have proposed lightweight architectures, optimization techniques, and edge-cloud frameworks to enhance detection performance without increasing computational demands. This section categorizes the most relevant studies into lightweight detection models, generative approaches, edge-based optimizations, performance improvements under challenging conditions, and small-object detection strategies.

### 2.1. Lightweight Object Detection Models

One-stage detectors like YOLO are extensively used in autonomous driving for their computational efficiency and real-time performance. Numerous lightweight variants have been proposed to enhance YOLO’s efficiency for deployment on edge devices with limited resources.

Zhou et al. [15] introduced MobileYOLO, which integrates MobileNetV2 and Efficient Channel Attention (ECA) to minimize computational complexity without sacrificing accuracy. Their model achieved 90.7% accuracy on the KITTI dataset, with an 80% reduction in model size compared to YOLOv4, enhancing its applicability in real-time edge computing scenarios. Similarly, ShuffYOLOX replaced CSPDarkNet53 with a ShuffleDet backbone, improving computational efficiency and achieving a 92.2% mAP, thus demonstrating its potential for autonomous navigation [20].

Yasir et al. [21] proposed SwinYOLOv7, which combines Swin transformers with YOLOv7 for robust ship detection in SAR imagery. Although it delivers cutting-edge performance, the use of transformer modules and anchor-free heads adds significant complexity. limiting its feasibility for edge deployment.

Yang and Fan [22] introduced YOLOv8-Lite, a lightweight adaptation of YOLOv8 optimized using the FastDet backbone, TFPN pyramid structure, and CBAM attention. Their model performed well on the NEXET and KITTI datasets, balancing speed and accuracy for real-time intelligent transportation applications. However, the reliance on transformer-style feature fusion (TFPN) and CBAM attention may still impose constraints under stricter edge limitations. Similarly, Wei et al. [23] developed SCCA-YOLO, integrating spatial and channel collaborative attention mechanisms to improve detection in wildlife monitoring. Yet, Performance was assessed on a custom dataset alongside the COCO128 subset, rather than on real-world urban driving benchmarks. While these attention-based designs offer practical enhancements, their applicability to real-time, street-level edge deployments remains to be validated. Recently, YOLOv11 [24] and YOLOv12 [25] models were introduced as the latest advancements in the YOLO family. YOLOv11 features modular improvements, including dynamic convolution aggregation and progressive feature refinement, targeting scalability and accuracy enhancements across multiple object detection tasks. YOLOv12 builds upon this by integrating R-ELAN and FlashAttention, enabling more efficient attention mechanisms and deeper feature propagation while maintaining high throughput. Both models include nano-scale variants, YOLOv11n and YOLOv12n, explicitly designed for real-time inference under resource constraints. These variants exemplify the trend toward ultra-lightweight detectors with optimized architectural efficiency, aligning closely with the objectives of our attention-based YOLOv5 enhancements.

Beyond structural modifications, recent studies have further explored integrating generative approaches to enhance object detection systems for autonomous driving. Although transformer-based detectors such as DETRs [26] have surpassed YOLO models’ accuracy, their high computational cost and latency hinder real-time use on edge devices. In contrast, our attention-enhanced YOLOv5 variants are explicitly optimized to trade off accuracy and efficiency, ensuring suitability for low-power embedded systems, latency-sensitive deployment scenarios.

### 2.2. Emerging Generative Approaches

Generative AI has recently been employed to enhance object detection and perception tasks, especially in autonomous systems. For instance, synthetic data generation using generative AI has improved robustness and generalization in complex driving environments [27,28]. Additionally, generative architectures such as GenCoder [29] have been proposed for anomaly detection in intra-vehicle networks, while multi-modal generative communication systems have been explored for intelligent vehicular ecosystems [30]. These trends indicate a growing synergy between generative and discriminative models.

Our work, however, focuses on optimizing discriminative YOLO-based models for low-latency use cases under edge constraints. For instance, Zhang et al. [31] proposed CAE-YOLOv5, which integrates CBAM and ECA into YOLOv5, improving detection accuracy to 96.3%. However, added attention layers increased inference time, limiting their suitability for edge deployment. Likewise, YOLOv5-NAM [32] integrates a Normalization-based Attention Module (NAM), improving small object detection and increasing mAP by 1.6% over the baseline while preserving real-time performance.

In deviation from earlier works focusing solely on lightweight design or attention-based accuracy gains, we propose three YOLOv5-based models—BaseECA, BaseECAx2, and BaseSE-ECA—each exploring distinct architectural trade-offs. BaseECA integrates ECA into the backbone to enhance channel-wise attention with minimal overhead. BaseECAx2 extends attention to the backbone and detection head for improved feature fusion. BaseSE-ECA combines SE attention in the backbone with ECA in the head to capture global and local contextual features. These models are thoroughly benchmarked across mAP, FPS, parameter count, and GFLOPs to validate their effectiveness in edge-constrained, real-time autonomous driving applications.

### 2.3. Edge-Based and Energy-Efficient Object Detection Approaches

Object detection models designed for edge deployment must prioritize energy efficiency and low computational cost. Liang et al. [19] proposed Edge YOLO, which offloads intensive computations to the cloud while performing lightweight inference on edge devices. Though effective, its dependency on persistent connectivity limits its suitability for fully offline systems.

Cai et al. [17] introduced a pruning-based version of YOLOv4, improving inference speed by 31.3% while maintaining competitive performance. Similarly, Wang et al. [33] proposed CRL-YOLOv5, which integrates Receptive Field Blocks (RFB) and CBAM to enhance small object detection by 5.4%. However, increased complexity and memory usage hinder their application on embedded devices.

While pruning, edge-cloud cooperation, and attention-based enhancements contribute to improved efficiency, achieving an optimal trade-off among accuracy, latency, and resource utilization remains a persistent challenge. Unlike generative or hybrid models, our attention-enhanced YOLOv5 variants are explicitly designed to offer a balanced solution, providing robust detection performance with minimal computational cost, making them more viable for real-time edge-based autonomous systems.

### 2.4. Object Detection Performance in Challenging Driving Environments

Autonomous driving systems must reliably detect objects under adverse conditions, such as occlusion, high-speed motion, and low lighting. Jia et al. [34] optimized YOLOv5 through structural reparameterization and Neural Architecture Search, achieving 96.1% accuracy and 202 FPS on the KITTI dataset. Liu et al. [35] introduced a lightweight model for traffic sign detection using Dense CSP modules, improving efficiency by 5.28%. Wang et al. [36] introduced YOLOv3-MT, which integrates Kalman filtering and DIoU-NMS for robust multi-object tracking.

Recent work has also improved detection in aerial and nighttime scenarios. Li et al. [37] proposed R-YOLOv5, which achieved 90.23% mAP with only 2.02M parameters on UAV datasets. Almujally et al. [38] enhanced nighttime surveillance using MIRNet-based low-light enhancement with YOLOv5, reaching 92.4% precision. While effective, these models are generally tailored for specialized environments. In contrast, our work focuses on versatile, general-purpose detection under various autonomous driving conditions. These studies underscore the critical role of attention and feature enhancement in improving detection robustness.

### 2.5. Small-Scale Object Detection in Autonomous Navigation

Detecting small objects is particularly challenging due to low resolution, occlusion, and background clutter. Various studies have applied super-resolution techniques and advanced attention mechanisms to mitigate these issues [39]. Zhao et al. [40] introduced SatDetX-YOLO, an improved YOLOv8-based model for detecting vehicles in satellite imagery. The model incorporates Deformable Attention Modules (DAM) and a Maximum Probabilistic Distance IoU (MPDIoU) loss, improving precision by 3.5% and recall by 3.3%. While tailored for satellite images, its focus on refined attention for small object detection aligns with our approach to enhancing YOLOv5’s performance in real-time, complex driving environments.

These studies emphasize the importance of advanced attention-based architectures and super-resolution techniques for enhancing small object detection in complex environments. Table 1 summarizes related works, detailing recent approaches’ methods, results, and key limitations. While many techniques demonstrate strong performance, they often suffer from increased computational costs or limited adaptability in real-world scenarios. Building on a comprehensive survey by Adam and Tapamo [41], which classifies deep learning-based vehicle detection methods and highlights the challenges of edge deployment, this study introduces optimized YOLOv5 variants incorporating SE and ECA attention modules. Unlike the survey’s broader synthesis, our work presents and evaluates concrete, lightweight models that balance accuracy and efficiency for real-time applications in autonomous driving.

## 3. Materials and Methods

This study introduces lightweight models for edge devices for lowering computational cost and improving affordability. Models based on YOLOv5s, a single-stage object detector, predict bounding boxes, class probabilities, and abjectness scores simultaneously. YOLOv5s is designed for real-time object detection, balancing inference speed with detection precision; its architecture is displayed in Figure 1. In this section, we provide a detailed explanation of the baseline model and the optimization techniques used to improve the baseline model, which resulted in the proposal of three lightweight models, namely: BaseECA, BaseECAx2, and BaseSE-ECA; their architectures are shown in Figure 2, Figure 3, and Figure 4, respectively. These models integrate ECA and SE mechanisms into the YOLOv5s architecture to improve channel-wise attention and feature selection. ECA and SE are lightweight mechanisms that will enhance the models’ focus on significant features by filtering out irrelevant details, reducing the models’ size, and improving their speed. ECA achieves this by adaptively recalibrating channel-wise feature responses without dimensionality reduction; SE leverages global average pooling combined with fully connected layers to capture channel dependencies.

### 3.1. YOLOv5

YOLOv5, which stands for You Only Look Once version five, is a highly optimized object detection model introduced by Ultralytics [42]; it is highly efficient and achieves cutting-edge results on the widely recognized Microsoft COCO benchmark [43], making it a widely adopted model for real-time object detection tasks owing to effectively balancing speed and accuracy. YOLOv5 is built upon three fundamental components that define its architecture: a backbone, a neck, and a head, as shown in Figure 1. The backbone is based on CSPDarknet53, an enhanced variant of Darknet53 used in YOLOv4 [44]. It integrates cross-stage partial (CSP) connections, which partition the feature maps into two separate segments, process one through a dense block, and concatenate the outputs, reducing computational costs and improving gradient flow. Additionally, YOLOv5 introduces a focus layer at the beginning of the backbone, which slices the input image into four parts and concatenates them, reducing spatial dimensions while preserving channel information to enhance efficiency. The neck in YOLOv5 employs Path Aggregation Network (PANet) to aggregate and enhance features extracted from different backbone levels. PANet extends Feature Pyramid Network (FPN) by incorporating a bottom-up path augmentation, which complements the top-down path of FPN and enhances object localization. Features from multiple levels are concatenated and refined through convolutional layers to improve multi-scale feature representation, ensuring robust detection across varying object sizes. The detection head generates the final predictions based on the YOLOv3 head [45]. It comprises three key components: bounding box predictions, objectness scores, and class probabilities. Bounding boxes are defined by their center point (x, y), as well as their width (w) and height (h), parameterized relative to grid cells and anchor boxes. The objectness score is a binary classification value that indicates whether a bounding box contains an object. At the same time, class probabilities represent a probability distribution over detected classes computed using sigmoid activation to support multi-label classification. The image undergoes initial processing in the input layer prior to being forwarded to the backbone for feature extraction. The backbone produces feature maps at multiple scales fused in the neck to produce three features: 80 × 80 resolution for detecting small objects, 40 × 40 resolution for medium objects, and 20 × 20 resolution for large objects. The output features are used in the head for object prediction, where confidence scores and bounding-box regression are performed using preset anchors. Finally, confidence thresholds and non-maximum suppression (NMS) are applied to filter out irrelevant detections and produce the final predictions.

### 3.2. Proposed Models

#### 3.2.1. BaseECA Model

Convolutional Neural Networks (CNNs) used in object detection must efficiently extract features while balancing accuracy and computational complexity. The YOLOv5 model employs cross-stage partial (CSP) bottleneck modules (C3 layers) in its backbone to capture hierarchical features at various levels. However, these C3 modules introduce a considerable number of parameters, which increases the computational burden. To address this challenge, we propose an optimized variant of YOLOv5, termed BaseECA, which integrates the Efficient Channel Attention (ECA) mechanism into the model’s backbone. As illustrated in Figure 2, we specifically replace the fourth C3 module, which is the last one in the backbone, positioned just before the Spatial Pyramid Pooling-Fast (SPPF) layer. We selected this layer because it refines high-level semantic features and contributes directly to all subsequent layers in the neck and head. Moreover, its output is concatenated with previous C3 outputs in the neck, making it a critical point for enhancement. ECA is applied to strengthen channel-wise feature representation by emphasizing the most informative channels without introducing significant additional complexity. This attention mechanism enables the network to prioritize essential features while reducing the overall parameter count and computational overhead. Although the original C3 module, composed of convolutional layers with CSP connections, is practical at high-level feature extraction, our ECA integration achieves a more efficient balance between performance and resource usage. The resulting BaseECA model maintains robust representational power while improving efficiency, making it more suitable for deployment in resource-constrained environments.

Parameter Complexity in the C3 Module: The C3 module in YOLOv5 performs convolutional operations while utilizing CSPNet’s partial feature reuse mechanism. It consists of two convolutional layers applied to half of the input channels and one applied to all channels, followed by batch normalization and non-linear activation functions (see Figure 2). The C3 component in layer nine introduces over 1.18 million trainable parameters, significantly contributing to the architecture’s computational complexity.The total number of trainable parameters in the C3 module can be estimated as(1)$${\mathrm{Params}}_{C3}=2\times \left(\frac{C}{2}\times \frac{C}{2}\times K^{2}\right)+\left(C\times C\times K^{2}\right)$$
where *C* represents the number of input and output channels, and *K* indicates the kernel size.Parameter Complexity in the ECA Module: The Efficient Channel Attention (ECA) module is an efficient module designed to optimize feature selection while maintaining computational efficiency. Compared to the C3 module, the ECA module applies a 1D convolution-based attention mechanism to refine channel-wise features.Replacing this modification reduces the parameter count from 1.18 million to just 1, enhancing model efficiency. This transformation lowers memory requirements and reduces computational cost (GFLOPs) while maintaining or improving feature refinement quality, demonstrating the effectiveness of ECA in optimizing YOLOv5’s backbone.

#### 3.2.2. BaseECAx2 Model

Feature improvement across multiple scales in deep learning models for object detection improves accuracy. However, how can we enhance feature representation while keeping computational costs minimal? To address this challenge, we introduce BaseECAx2, the architecture in Figure 3, an extension of the BaseECA architecture that further integrates the ECA module directly into the backbone architecture and the detection head. The motivation behind BaseECAx2 is to enhance both intermediate and final-stage feature representations by embedding attention mechanisms in deeper layers and prediction heads in critical areas where semantic richness and localization precision are most needed. This approach is beneficial in detecting objects of varying sizes under different environmental conditions. In BaseECAx2, layer nine’s backbone’s C3 module is replaced with ECA, allowing the model to refine extracted features early in the network by emphasizing the most informative channels. So we have chosen this specific layer because it lies in the deeper part of the backbone, where semantic features are richer, as mentioned in the previous section. Its outputs influence multiple subsequent layers in the neck and head. Thus, modifying it can significantly enhance downstream predictions. An additional ECA module is positioned after the C3 module in the detection head, enhancing the feature maps used for final predictions. At this stage, the detection head operates at three resolutions, 80 × 80, 40 × 40, and 20 × 20, and ECA enables the model to apply adaptive channel attention at multiple scales, which helps the model better capture fine-grained details and large-scale context, improving object localization and classification precision. To analyze the computational impact, consider the parameter complexity of ECA. Unlike the C3 module, which consists of multiple convolutional layers, ECA is lightweight and only requires a 1D convolution with an adaptive kernel size *k* given by(2)$${\mathrm{Parameters}}_{\mathrm{ECA}}=C\times K$$
where *C* and *K* are as previously defined, by replacing C3 in the backbone and inserting ECA in the head, BaseECAx2 minimizes the model’s parameters and computational overhead while sustaining or enhancing detection performance. Therefore, it ensures that the feature maps are refined before being fed into the neck and head, improving the feature discrimination across scales and resulting in a model that successfully balances performance and computational efficiency, making BaseECAx2 a powerful solution for real-time object detection.

#### 3.2.3. BaseSE-ECA

While BaseECAx2 improves multi-scale feature selection by incorporating Efficient Channel Attention (ECA) in both the backbone and detection head, one question remains: Can we further refine feature representation to improve detection precision? To address this, we introduce BaseSE-ECA, which combines the strengths of Squeeze-and-Excitation (SE) and ECA modules to optimize feature extraction and refinement. BaseSE-ECA enhances the model’s ability to model complex channel relationships in the backbone while preserving ECA’s lightweight and adaptive properties in the detection head. Our dual-attention approach aims to refine the deep semantic and final prediction-level features, ensuring higher precision in diverse detection scenarios. BaseSE-ECA was trained and evaluated in two configurations: one using all object classes in the KITTI dataset, and another trained exclusively on the vehicle class, which we refer to as BaseSE-ECA (vehicles). This class-specific variant demonstrated superior performance in detecting vehicles with high precision, highlighting the model’s ability to specialize effectively when attention mechanisms are strategically applied. Building upon BaseECAx2, BaseSE-ECA modifies the backbone by replacing the final C3 module with an SE module while keeping the ECA module in the detection head, as illustrated in Figure 4. As mentioned in the above sections, we chose the last C3 module in the backbone, crucial in summarizing deep semantic information for the neck and head. The SE module strengthens channel-wise feature representation by modeling interdependencies between channels, allowing the model to refine its feature selection. The ECA module in the head ensures that the final feature maps used for detection recalibrate optimally, improving classification and localization accuracy. This strategic combination enables the model to benefit from SE’s ability to capture complex channel dependencies and ECA’s computational efficiency in attention refinement. The Efficient Channel Attention (ECA) module applies a lightweight 1D convolution-based mechanism to enhance channel-wise features without dimensionality reduction. Unlike SE, which introduces fully connected layers, ECA maintains spatial information while recalibrating the importance of features. Mathematically, the ECA operation can be calculated in the following equation:(3)$$A_{\mathrm{ECA}}\left(x\right)=\sigma \left(\mathrm{Conv}1D\left(g\left(x\right)\right)\right)$$
where g(x) denotes global average pooling across the spatial dimensions; Conv1D uses a small, dynamically selected kernel size *K* to model local channel interactions; and σ represents the sigmoid activation function responsible for scaling the attention weights. On the other hand, the Squeeze-and-Excitation (SE) module follows a two-step process: the first step is squeeze, which uses global average pooling (GAP) to compress spatial dimensions into a channel descriptor. The second is excitation, which includes two fully connected layers that apply transformations to model channel dependencies. Mathematically, the SE operation is given by(4)$$A_{\mathrm{SE}}\left(x\right)=\sigma \left(W_{2}\delta \left(W_{1}g\left(x\right)\right)\right)$$
where $W_{1}$ and $W_{2}$ refer to the trainable weights of the two fully connected layers, δ denotes the ReLU activation function, and σ is the sigmoid activation function responsible for rescaling the feature responses.

## 4. Experiments

### 4.1. Dataset and Preprocessing

The KITTI dataset stands out as one of the leading benchmark datasets to evaluate computer vision algorithms in autonomous driving scenarios. It provides diverse real-world driving scenes captured using high-resolution stereo cameras and 3D LiDAR sensors. The dataset includes various levels of occlusion and truncation, making it well-suited for testing the robustness of lightweight object detection models under challenging conditions.The dataset is partitioned into three distinct subsets: for training, 5220 images; for validation, 1495 images; and for testing, 746 test images. It includes eight object categories, covering common road participants: car, van, truck, pedestrian, person (sitting), cyclist, tram, and miscellaneous. The dataset contains 40,484 labeled objects, averaging 5.4 annotations per image across these eight classes.Data preprocessing was conducted prior to training to ensure a consistent input structure with the proposed detection models and improve training efficiency. All images were scaled to a 640 × 640 pixel resolution while maintaining the aspect ratio using stretching techniques. Input pixel values were normalized to a range of [0,1] to ensure stable convergence during training. Data augmentation techniques, including random horizontal flipping, brightness adjustment, and affine transformations, were incorporated to improve generalization and model robustness [46].The KITTI dataset served as the primary evaluation benchmark in this study due to its well-structured annotations and established reputation for testing object detection models in autonomous driving scenarios. The proposed models achieved strong performance on KITTI, which was used as a reference to measure the models’ effectiveness in standard conditions. See some samples for detection results in Figure 5.The BDD-100K Dataset is a comprehensive driving video dataset tailored for autonomous driving research, featuring 100,000 samples of annotated video clips collected from diverse geographic locations and environmental conditions, including urban streets, highways, and residential areas. The dataset features weather conditions such as explicit, cloudy, rainy, and foggy scenarios and daytime and nighttime driving scenes, making it suitable for evaluating model robustness in complex environments [47]. The BDD-100K dataset was used to explore the limitations of the proposed models under more complex conditions. See some samples for detection results in Figure 6. While the models demonstrated strong performance on KITTI, they exhibited reduced performance on BDD-100K, particularly in scenes with poor lighting, motion blur, and occlusions. This evaluation provided insights into the models’ robustness and highlighted areas for future improvements, such as enhanced temporal feature fusion and improved noise handling techniques.

### 4.2. Implementation and Training

We implemented all three models (BaseECA, BaseECAx2, and BaseSE-ECA) using the PyTorch framework (version 2.0.1+cu117) and trained them on the KITTI dataset. We conducted the training over 300 epochs on an NVIDIA Quadro P600 GPU (4 GB VRAM, embedded in a Dell laptop; NVIDIA Corporation, Santa Clara, CA, USA), utilizing CUDA 11.7 for acceleration. The experiments were run on a system with Python 3.11.4 and PyTorch 2.0.1, operating on Ubuntu 24.04 LTS, equipped with an Intel Core i7-8850H CPU (Intel Corporation, Santa Clara, CA, USA) and 32 GB of RAM. We adopted the default hyperparameters from the official YOLOv5s implementation, as the developers empirically optimized them for object detection tasks. Specifically, we used a batch size of 8, an initial learning rate of 0.01, and the SGD optimizer with a momentum of 0.937 and weight decay of 0.0005. We applied a one-cycle learning rate scheduler with initial and final learning rates set to 0.01 to accelerate convergence. To enhance model generalization, we applied a series of data augmentation techniques, including mosaic augmentation (probability 1.0), random horizontal flipping (probability 0.5), hue adjustment (±0.015), saturation adjustment (±0.7), brightness adjustment (±0.4), image scaling (±0.5), and image translation (±0.1). We also experimented with alternative hyperparameter settings, such as modifying the learning rate, batch size, and augmentation configurations, but none improved performance or training stability. Based on these observations, we retained the default YOLOv5 configuration, as it consistently yielded the best balance between convergence speed, robustness, and detection accuracy.

### 4.3. Performance Metrics

To thoroughly evaluate the effectiveness of the proposed models, we used key performance metrics that assess both inference speed and detection accuracy. These include frames per second (FPS) for speed evaluation and mean average precision (mAP) for accuracy evaluation. Precision and recall were also used to assess detection quality.

#### 4.3.1. Inference Speed

The inference speed was evaluated using frames per second (FPS), reflecting the image throughput per second. Higher FPS values indicate faster inference and better real-time performance, crucial for deployment in autonomous driving systems. FPS is calculated as follows:(5)$$\mathrm{FPS}=\frac{N}{T}$$
where *N* = total number of processed images, and *T* = total inference time in seconds.

#### 4.3.2. Detection Accuracy

Detection accuracy was assessed using mean average precision (mAP), which evaluates the model’s precision across all object classes in the dataset. Average precision (AP) for a single class is calculated as the area under the precision–recall (*P*–*R* ) curve:(6)$$\mathrm{AP}=\int_{0}^{1}P\left(R\right)\;dR$$
where P(R) = precision as a recall function.

mAP is obtained by averaging the AP scores across all *C* classes:(7)$$\mathrm{mAP}=\frac{1}{C}\sum_{i=1}^{C}{\mathrm{AP}}_{i}$$
where *C* = total number of object classes, ${\mathrm{AP}}_{i}$ = AP for the *i*-th class.

Higher mAP values indicate better overall detection accuracy across multiple object classes.

#### 4.3.3. Detection Quality Assessment

We used precision and recall metrics to evaluate further detection quality, which provides an idea of the model’s ability to identify positive detections while correctly mitigating false positives and negatives.(8)$$\mathrm{Precision}=\frac{\mathrm{TP}}{\mathrm{TP}+\mathrm{FP}}$$(9)$$\mathrm{recall}=\frac{\mathrm{TP}}{\mathrm{TP}+\mathrm{FN}}$$
where TP (true positives) refers to correct detection, FP (false positives) are incorrect detections, and FN (false negatives) are undetected objects. A higher precision value reflects the ability of a model to minimize false positives, while a high recall value indicates improved detection of all relevant objects. Balancing these metrics ensures optimal performance for both accuracy and reliability in real-world autonomous driving scenarios. Combining these evaluation metrics assesses the proposed models’ efficiency, accuracy, and practical feasibility for real-time deployment on edge devices.

### 4.4. Ablation Study

Object detection models must efficiently extract, refine, and prioritize features while balancing accuracy and computational cost. We conducted an ablation study comparing the standard C3 block, SE attention, and ECA to explore how different architectural modifications affect performance. The goal was to understand how replacing C3 with SE or ECA impacts feature importance, computational efficiency, accuracy, and speed. The architectural characteristics of the C3, SE, and ECA modules are described in Table 2, while their quantitative performance comparisons are detailed in Table 3 and Table 4, highlighting the balance between accuracy, inference speed, and computational cost.

#### 4.4.1. Impact on Feature Importance

The standard C3 block in YOLOv5s architecture processes all channels equally, treating each feature with the same level of importance, which in some scenarios might lead to redundant or less informative features being processed. In contrast, SE and ECA modules prioritize the most relevant features by introducing attention mechanisms. While SE explicitly models channel dependencies using fully connected (FC) layers, ECA applies a lightweight 1D convolution, allowing it to capture local channel interactions efficiently.

#### 4.4.2. Computational Cost Considerations

Replacing C3 with SE attention reduces the number of convolution operations. However, it introduces additional costs from fully connected layers used to model channel dependencies, making SE computationally lighter than C3 but still more expensive than ECA. On the other hand, ECA replaces the FC layers with a more efficient 1D convolution, significantly reducing overhead while maintaining strong feature representation, which makes ECA the most computationally efficient option among the three.

#### 4.4.3. Effect on Accuracy

Since SE and ECA refine feature selection, they both contribute to accuracy improvements over the standard C3 module. SE enhances accuracy by globally analyzing channel dependencies, making it beneficial for detecting large objects or those with distinct characteristics. However, ECA refines attention at a more localized level, improving object recognition in scenarios where subtle local variations matter, making ECA particularly effective in handling complex backgrounds and smaller objects.

#### 4.4.4. Effect on Speed

Inference speed plays a crucial role in real-time object detection tasks, and the standard C3 block exhibits slightly slower performance due to its increased number of convolution operations. SE attention speeds up inference by reducing convolutional complexity but is slowed down by fully connected layers. ECA achieves the best trade-off, as its 1D convolution operation is significantly lighter than SE’s fully connected layers, making it the fastest option while improving feature representation.

These tables show that ECA offers the most favorable balance among the three modules. It delivers the highest mAP and FPS with reduced computational load, confirming its suitability for real-time edge deployment.

## 5. Results

The proposed models were evaluated against the YOLOv5 baseline to assess the impact of Squeeze-and-Excitation and Efficient Channel Attention modules on detection accuracy, computational efficiency, and inference speed. The evaluation was conducted using precision (P), recall (R), mean average precision at 50% (mAP@50), mean average precision at 95% (mAP@95), floating-point operations (GFLOPs), parameter count, model size, and inference speed (FPS). A comprehensive comparative analysis of detection performance, computational efficiency, and class-wise detection accuracy is presented in Table 3, Table 4 and Table 5.

### 5.1. Detection Performance Analysis

Table 3 provides the proposed models’ precision, recall, and mAP scores and the baseline YOLOv5s. The results demonstrate that integrating attention mechanisms in the baseline model impacts detection accuracy, with variations based on the placement of ECA and SE modules.

The following observations can be drawn from Table 3:BaseSE-ECA (vehicles) achieves the highest detection precision (96.7%) and recall (96.2%), demonstrating that attention mechanisms significantly enhance vehicle detection performance.BaseECA surpasses the baseline model in mAP@50 (90.9%) and mAP@95 (68.9%), indicating improved overall detection accuracy.BaseECAx2, incorporating dual ECA, demonstrates robust detection capabilities with precision (89.1%) and recall (86%), highlighting the effectiveness of multi-layer attention integration.While BaseSE-ECA achieves the highest precision (91.3%), its lower recall (83.9%) suggests a trade-off between high selectivity and overall detection capability.

These findings indicate that the strategic placement of SE and ECA modules significantly influences model performance, with SE improving large-object detection and ECA enhancing small-object recognition.

While the proposed models performed well on the KITTI dataset, they exhibited noticeable performance degradation on the BDD-100K dataset. This decline is primarily attributed to the dataset’s complex driving scenarios involving low lighting, motion blur, and occlusions. Table 6 presents the overall detection metrics on the BDD-100K dataset, while Table 7 provides a class-wise breakdown to analyze this degradation in greater detail.

The following observations can be drawn from Table 7:Car detection remains consistently high across all models (≥75%), confirming model robustness for prominent and frequently occurring object classes in driving scenes.Motorcycle and rider classes show the sharpest degradation, with mAP@50 ranging between 21.1%–25.1% for “Motor" and 31.9%–35.6% for “Rider”, indicating difficulty in detecting fast-moving, small, or partially occluded objects in real-world conditions.Bike and person detection performances are modest across models, with mAP below 51%. Detection challenges likely stem from partial visibility, variable human poses, and low-light conditions.Bus and truck classes perform relatively well due to their larger sizes and clearer boundaries. BaseSE-ECA achieves the best result on “Bus” (50.1%) and competitive accuracy on “Truck” (53.4%), demonstrating that attention mechanisms help in large-object detection under complex conditions.

### 5.2. Computational Efficiency and Model Complexity

Table 4 shows the comparative analysis of GFLOPs, parameter count, model size, and inference speed (FPS). The results show the computational benefits of integrating ECA and SE modules, achieving an optimal trade-off between detection accuracy and computational efficiency. The key findings are

BaseSE-ECA (vehicles) achieves a compact model size (9.1 MB) with a moderate computational complexity of 13.7 GFLOPs, making it highly efficient for edge-device deployment.BaseECAx2 exhibits the lowest GFLOPs (13.0) while maintaining fewer parameters (43.7 M ), confirming that multi-layer ECA integration reduces computational overhead.BaseECA, despite enhancing feature representation, slightly increases computational complexity, requiring 14.9 GFLOPs and a model size of 12.1 MB.The baseline YOLOv5s model has the highest GFLOPs (15.8) and parameter count (70 M) while achieving the lowest FPS (34), reinforcing the efficiency gains of the proposed models.

These results confirm that ECA-based modifications effectively reduce model complexity, improving computational efficiency for real-time edge applications; Figure 7 shows the proposed models with respect to model size, number of parameters, computational cost, and speed.

### 5.3. Class-Wise Detection Performance

In Table 5, the results show different object categories measured on the mAP@50, illustrating the strengths of each model in detecting specific classes.

BaseECA achieves the highest pedestrian detection accuracy (85.7%) and performs exceptionally well for cyclists (91.4%), confirming that ECA enhances small-object detection.BaseSE-ECA (vehicles only) excels in vehicle detection, achieving 98.8% mAP for vans and 98.4% for trucks, reinforcing SE’s strength in large-object recognition.BaseECAx2 maintains strong multi-class performance, demonstrating a balanced detection capability across different object categories.

These findings demonstrated that attention-based modifications improve class-specific detection accuracy, particularly for human and vehicle detection, which are critical for autonomous driving applications. Figure 8 shows the performance of each model on all classes.

### 5.4. Discussion

Deploying deep learning-based object detectors on edge devices remains a significant challenge due to strict constraints on memory, processing power, and energy consumption. To address this, we proposed lightweight variants of YOLOv5 by integrating Efficient Channel Attention (ECA) and Squeeze-and-Excitation (SE) modules, aiming to enhance feature extraction with minimal computational overhead. Our findings demonstrate an effective balance between detection accuracy and computational efficiency. For example, the BaseECAx2 model achieved a mAP@50 of 90.0% with only 4.37 M parameters and 13.0 GFLOPs, outperforming traditional models like YOLOv3 and RetinaNet regarding resource efficiency. Additionally, the BaseSE-ECA model achieved high precision (91.3%) and robust multi-class detection performance, as shown in Table 6 and Table 8. Unlike previous studies such as CAE-YOLOv5 [31] and CRL-YOLOv5 [33], which apply attention mechanisms exclusively within the backbone, our study introduces a dual-placement strategy. Specifically, BaseECAx2 applies ECA modules in both the backbone and detection head, while BaseSE-ECA combines SE in the backbone with ECA in the head. Our models achieve up to a 3× reduction in computational cost compared to recent attention-enhanced models such as SNCE-YOLO (35 GFLOPs) and YOLOv4-5D (103 GFLOPs) while maintaining comparable or superior accuracy. Notably, the smallest model size is just 9.1 MB, with real-time inference speeds, making the designs well-suited for deployment on resource-constrained edge devices. We also compared our models to YOLOv8-Lite [22], a recent low-resource design. While YOLOv8-Lite achieved 76.62% mAP@50 on the KITTI dataset using 4.8M parameters and 8.95 GFLOPs, our BaseECAx2 achieved a substantially higher mAP@50 of 90.0% with a similar number of parameters and only moderately higher complexity, demonstrating a superior accuracy–efficiency trade-off for practical edge applications. On the BDD-100K dataset, which includes more challenging scenarios such as occlusion, low light, and motion blur, our models experienced a moderate drop in performance compared to KITTI; this suggests that while channel-wise attention mechanisms (SE, ECA) improve spatial feature learning, they are less effective in capturing spatiotemporal variations inherent in real-world driving conditions. Compared with other recent lightweight detectors, YOLOv11n [24] achieved a mAP@50 of 88.8% using 2.58M parameters and 6.3 GFLOPs. Our BaseECAx2 improved upon this by achieving 90.0% mAP@50 (+1.2%), with higher precision (89.1% vs. 85.2%) and a competitive inference speed of 40.0 FPS. While slightly larger, BaseECAx2 offers better detection reliability without sacrificing real-time performance. YOLOv12n [25], on the other hand, achieved a higher mAP@50 of 91.6% with 2.56 M parameters and 6.3 GFLOPs. However, it exhibited lower precision (87.6%) and slower inference (38.3 FPS) compared to our BaseECAx2 and BaseSE-ECA models. These results suggest that although YOLOv12n performs well in overall detection and recall, our models offer superior precision and faster execution—critical traits in safety-critical, real-time edge AI applications. In summary, the proposed architectural enhancements to YOLOv5 introduce effective and novel attention configurations that significantly improve the trade-off between accuracy, speed, and compactness. These contributions make our models practical and deployable solutions for real-time edge AI applications, including autonomous vehicles, intelligent surveillance, and embedded vision systems.

### 5.5. Future Work

Future work will focus on enhancing the proposed models to handle challenging scenes better, similar to those found in the BDD-100K dataset, which include low-light environments, motion blur, and heavy occlusions. We plan to integrate temporal modeling techniques such as ConvLSTM-based feature fusion and transformer-style multi-frame attention to better capture motion and temporal context. We also aim to apply domain adaptation methods—such as style transfer and test-time training—to simulate and generalize across adverse weather and lighting conditions. Furthermore, we will design controlled experiments to evaluate model robustness under sensor noise and environmental variability and validate our approach on complex real-world datasets such as Cityscapes and nuScenes.

## 6. Conclusions

In this study, we developed lightweight object detection models for real-time use on edge devices, such as those found in autonomous vehicles. Although the original YOLOv5 generally performs well, its high computational cost and large model size make it less suitable for environments with limited resources. To address this, we introduced three optimized variants by integrating Efficient Channel Attention (ECA) and Squeeze-and-Excitation (SE) modules at different levels of the YOLOv5 architecture. Experimental evaluations on the KITTI dataset demonstrated that the BaseECAx2 model achieves the best trade-off between detection accuracy and efficiency, achieving only 13 GFLOPs, a compact model size of 9.1 MB, and a fast inference speed of 27 ms, making it highly suitable for real-time edge deployment. For applications prioritizing detection accuracy, such as vehicle-specific recognition in autonomous systems, the BaseSE-ECA model delivered the highest performance, achieving 96.69% precision, 96.20% recall, and 98.40% mAP@50. Despite these promising results on KITTI, performance degraded on the more complex BDD-100K dataset due to challenges such as occlusion, low lighting, and motion blur. The results emphasize the need for further architectural enhancements to improve robustness under real-world driving conditions. This work introduces a practical and scalable direction for building efficient, high-performing object detectors using lightweight attention modules. The study demonstrates that improvements can effectively bridge the gap between computational efficiency and real-time performance, supporting the advancement of AI-driven perception in autonomous vehicles and other embedded vision applications.

## Figures and Tables

**Figure 1 jimaging-11-00263-f001:**
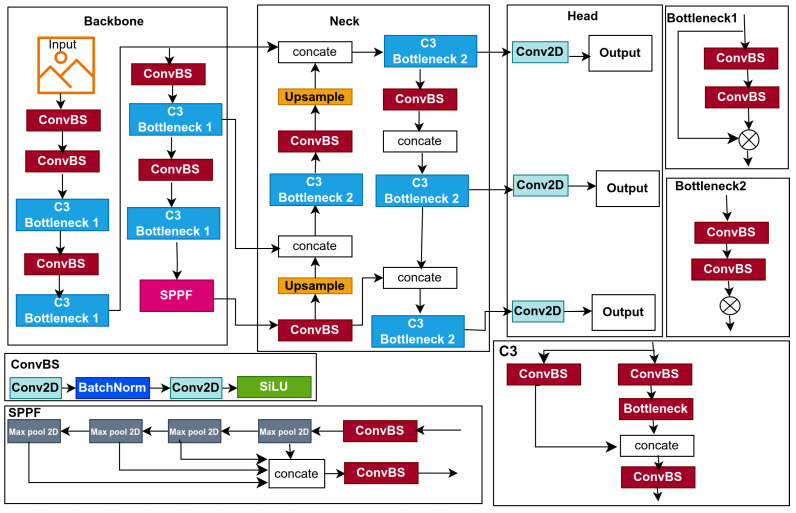
YOLOv5 network architecture in detail.

**Figure 2 jimaging-11-00263-f002:**
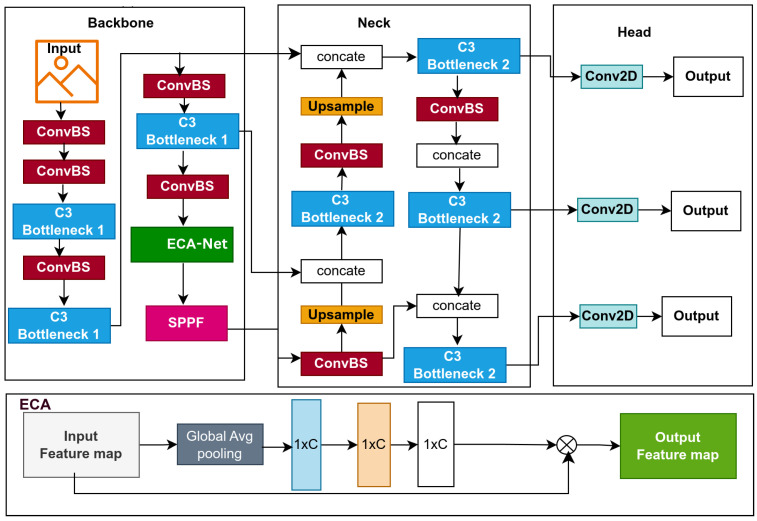
The proposed BaseECA network architecture and ECA architecture.

**Figure 3 jimaging-11-00263-f003:**
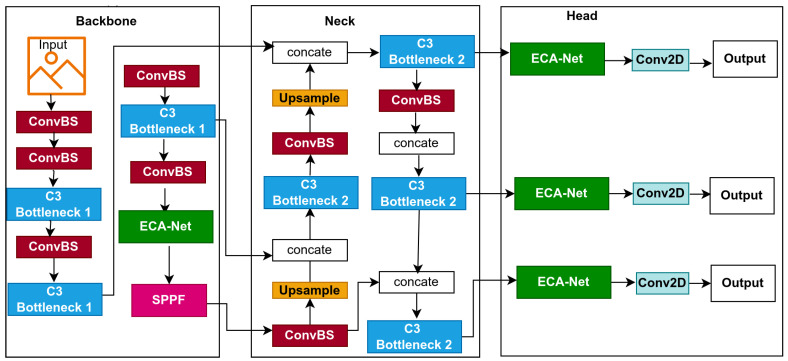
The proposed BaseECAx2 network architecture.

**Figure 4 jimaging-11-00263-f004:**
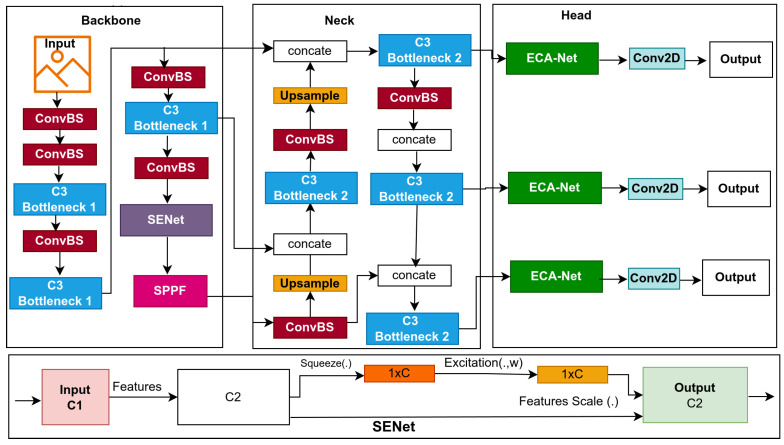
The proposed BaseSE-ECA network architecture.

**Figure 5 jimaging-11-00263-f005:**
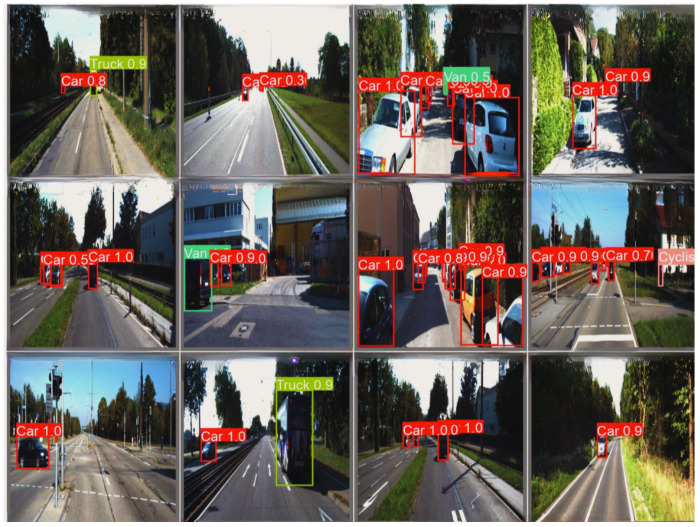
Detection result of BaseECAx2 on the KITTI dataset under various conditions.

**Figure 6 jimaging-11-00263-f006:**
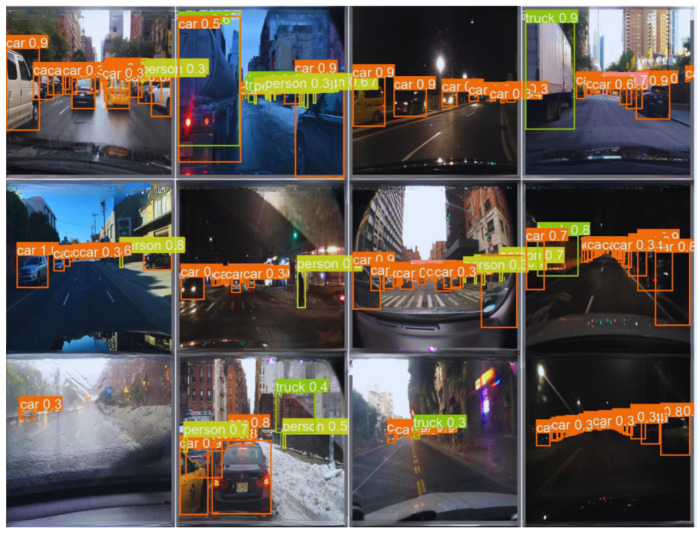
Detection result of BaseECAx2 on the BDD-100K dataset under various conditions.

**Figure 7 jimaging-11-00263-f007:**
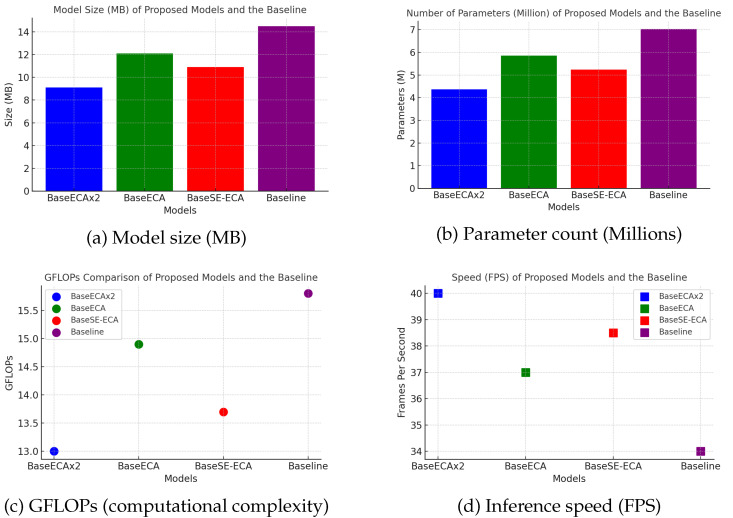
Comparison of the proposed models and the baseline across key performance metrics: (**a**) model size in megabytes (MB); (**b**) total number of parameters in millions; (**c**) computational cost measured in GFLOPs; (**d**) inference speed in frames per second (FPS).

**Figure 8 jimaging-11-00263-f008:**
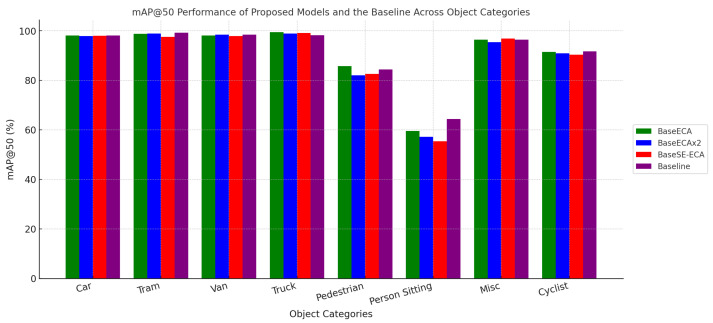
The comprehensive performance of the models across all classes concerning mean average precision (mAP).

**Table 1 jimaging-11-00263-t001:** Summary of recent object detection research highlighting methods, outcomes, and limitations.

Authors	Methods	Results	Limitations
Zhou et al. [15]	Combines MobileNetV2 and ECA.	90.7% accuracy, 80% model size reduction.	Struggles with occlusions and small objects.
Cai et al. [17]	Network pruning and feature fusion in YOLOv4.	31.3% faster inference with high accuracy.	Pruning requires careful tuning.
Liang et al. [19]	Edge-cloud offloading scheme.	Reduced latency and CPU load.	Requires a stable network for full functionality.
He et al. [20]	Replaces CSPDarkNet53 with ShuffleDet.	92.2% mAP with reduced complexity.	Weak performance in low-light scenes.
Zhang et al. [31]	Integrates ECA and CBAM into YOLOv5.	Achieved 96.3% detection accuracy.	Increased inference time due to added attention layers.
Wang et al. [32]	Incorporates NAM into YOLOv5.	+1.6% mAP vs. baseline.	Limited gain in dense traffic.
Jia et al. [34]	NAS and structural reparameterization.	96.1% accuracy, 202 FPS on KITTI.	Dataset-specific tuning limits generalization.
Liu and Dongye [35]	Apply dense CSP and enhanced FPN to YOLOv5.	5.28% boost in detection efficiency.	Lower performance in adverse weather conditions.
Wang et al. [36]	Kalman filtering + DIoU-NMS.	Up to 4.65% mAP increase.	Degraded under severe occlusion.
Zhao et al. [40]	Uses deformable attention and MPDIoU loss.	+3.5% precision, +3.3% recall in remote sensing.	High computational complexity.

**Table 2 jimaging-11-00263-t002:** Comparison of standard C3 block, SE attention, and ECA.

Method	Description
C3 block	Equal channel processing; more convolution operations; baseline accuracy; slightly slower speed.
SE attention	Focuses on key channels; introduces fully connected (FC) layer cost; improves global feature focus; faster than C3.
ECA attention	Local channel attention via 1D convolution; lightweight; boosts local feature refinement; fastest option.

**Table 3 jimaging-11-00263-t003:** Detection performance of proposed models and baseline on the KITTI dataset.

Model	Precision (%)	Recall (%)	mAP@50 (%)	mAP@95 (%)
BaseECA	88.0	87.5	90.9	68.9
BaseECAx2	89.1	86.0	90.0	65.9
BaseSE	91.3	83.9	89.7	67.6
BaseSE-ECA	91.3	83.9	89.9	67.6
BaseSE-ECA (Vehicle)	96.7	96.2	98.4	84.4
Baseline	89.1	87.9	91.4	69.5
YOLOv11n	85.2	82.8	88.8	34.9
YOLOv12n	87.6	89.2	91.6	71.9

**Table 4 jimaging-11-00263-t004:** Model comparison based on GFLOPs, parameter count, inference speed, model size, and improvement summary.

Model	GFLOPs	Params (M)	FPS	Size (MB)	Improvement Summary
BaseECA	14.9	5.85	37.0	12.1	+3.0 FPS, −0.9 GFLOPs, −2.4 MB
BaseECAx2	13.0	4.37	40.0	9.1	+6.0 FPS, −2.8 GFLOPs, −5.4 MB
BaseSE	15.6	6.71	35.0	13.8	+1.0 FPS, −0.2 GFLOPs, −0.7 MB
BaseSE−ECA	13.7	5.24	38.5	10.9	+4.5 FPS, −2.1 GFLOPs, −3.6 MB
BaseSE−ECA (Vehicles)	13.7	5.24	38.5	9.1	Same FPS, −2.1 GFLOPs, −5.4 MB
Baseline	15.8	7.03	34.0	14.5	Reference point
YOLOv11n	6.3	2.58	53.0	5.5	+19.0 FPS, −9.5 GFLOPs, −9.0 MB
YOLOv12n	6.3	2.56	38.3	5.6	+3.3.0 FPS, −9.5 GFLOPs, −8.9.0 MB

**Table 5 jimaging-11-00263-t005:** Comparison of mean average precision (mAP) performance for various models across all classes (%).

Model	Car	Tram	Van	Truck	Pedestrian	Person Sitting	Misc	Cyclist
BaseECA	98.1	98.8	98.1	99.4	85.7	59.5	96.4	91.4
BaseECAx2	97.8	98.9	98.4	98.9	82.0	57.2	95.4	90.9
BaseSE	98.1	99.0	98.5	99.2	84.6	67.4	96.5	91.0
BaseSE-ECA	98.0	97.5	97.9	99.1	82.6	55.4	96.8	90.3
BaseSE-ECA (Vehicles)	98.0	N/A	98.8	98.4	N/A	N/A	N/A	N/A
Baseline	98.1	99.2	98.4	98.2	84.4	64.4	96.4	91.7
YOLOv11n	97.3	94.9	96.1	97.2	80.6	66.6	80.0	82.7
YOLOv12n	98.2	98.1	98.3	98.4	85.2	68.8	94.3	91.1

**Table 6 jimaging-11-00263-t006:** Detection performance of proposed models and baseline on the BDD-100K dataset (updated after rerunning the test task).

Model	Precision (%)	Recall (%)	mAP@50 (%)
BaseECAx2	62.1	43.8	46.6
BaseSE	63.8	42.7	47.2
BaseSE-ECA	61.4	44.1	48.0
Baseline	57.5	44.4	47.0

**Table 7 jimaging-11-00263-t007:** Class-wise mAP@50 (%) performance of proposed models and baseline on the BDD-100K dataset.

Model	Bike	Bus	Car	Motor	Person	Rider	Truck
BaseECAx2	41.9	45.5	76.0	24.5	49.6	34.4	54.5
BaseSE	48.8	47.5	76.2	21.1	50.1	31.9	55.1
BaseSE-ECA	45.9	50.1	75.9	25.1	49.6	35.6	53.4
Baseline	46.1	48.7	76.3	21.2	51.0	32.2	53.4

**Table 8 jimaging-11-00263-t008:** Comparison of detection accuracy, model size, and computational complexity with state-of-the-art detectors on the KITTI dataset.

Method	mAP (%)	# Parameters (M)	GFLOPs
YOLOX-L [34]	92.27	54.15	155.69
ShuffYOLOX [34]	92.20	35.43	89.99
SNCE-YOLO [4]	91.90	9.58	35.20
MobileYOLO [15]	90.70	12.25	46.70
YOLOv8s [4]	89.40	11.13	28.40
SSIGAN and GCAFormer [5]	89.12	N/A	N/A
YOLOv4-5D[P-L] [17]	87.02	N/A	103.66
YOLOv3 [34]	87.37	61.53	N/A
SD-YOLO-AWDNet [12]	86.00	3.70	8.30
YOLOv6s [4]	85.60	16.30	44.00
YOLOv7s-tiny [12]	84.12	6.20	5.80
YOLOv3-MT [36]	84.03	N/A	32.06
RetinaNet [15]	88.70	37.23	165.40
YOLOv3 [48]	87.40	61.53	234.70
YOLOv8-Lite [22]	76.62	4.80	8.95
Faster R-CNN [36]	71.86	N/A	7.04
Edge YOLO [19]	72.60	24.48	9.97
MobileNetv3 SSD [19]	71.80	33.11	12.52
YOLOv4-5D[P-G] [48]	69.84	N/A	76.90
SSD [36]	61.42	N/A	27.06
S-DAYOLO [49]	49.30	9.35	19.70
YOLOv11n	88.8	2.60	6.30
YOLOv5s	91.4	7.03	15.8
Ours (BaseECA)	90.90	5.85	14.90
Ours (BaseECAx2)	90.00	4.37	13.00
Ours (BaseSE-ECA)	89.90	5.24	13.70
Ours (BaseSE-ECA (Vehicles))	98.40	5.24	13.70

## Data Availability

The datasets used in this study are publicly available. The KITTI dataset is available at http://www.cvlibs.net/datasets/kitti/ (accessed on 7 June 2025), and the BDD100K dataset is available at http://bdd-data.berkeley.edu/ (accessed on 10 June 2025). No new datasets were generated during the study.

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
