# Peer review of "Enhancing YOLOv5 for Autonomous Driving: Efficient Attention-Based Object Detection on Edge Devices"

_2313-433X, 2025, doi:10.3390/jimaging11080263_

Round 1
Reviewer 1 Report
Comments and Suggestions for Authors
What specific improvements should the authors consider regarding the
methodology?
Specific improvements the authors could consider regarding the methodology include designing experiments to systematically test the models under adverse weather conditions, or sensor noise (not obligatory).
Any additional comments on the tables and figures.
The tables are well-organized, making it easy to compare key metrics like precision, recall, mAP, and GFLOPs. Figures illustrating model architecture and attention module integration help readers understand the proposed modifications effectively. Enlarging Figures 7 and 8 will help improve their visibility and make it easier for readers to understand the details.
Author Response
Comment 1: What specific improvements should the authors consider regarding the
Methodology? Specific improvements the authors could consider regarding the methodology include designing experiments to systematically test the models under adverse weather conditions, or sensor noise (not obligatory).
Response 1:
We appreciate the reviewer’s insightful suggestion. Although assessing the models under adverse weather and sensor noise conditions is not included in the current scope of our research study, we agree that such experiments would provide valuable insights into model robustness. In the future works section, we have mentioned this as a potential direction in the revised Future Work, highlighting its importance for real-world deployment in autonomous driving systems.
Comment 2: The tables are well-organized, making it easy to compare key metrics like precision, recall, mAP, and GFLOPs. Figures illustrating model architecture and attention module integration help readers understand the proposed modifications effectively. Enlarging Figures 7 and 8 will help improve their visibility and make it easier for readers to understand the details.
Response 2:
Thank you for your positive feedback on the organization of the tables and figures. Regarding the reviewer’s suggestion, we have enlarged Figures 7 and 8 to improve their visibility and significantly enhanced their layout. Also, we have provided clear captions and annotations to facilitate better readability and effectively highlight the detection results under various environmental conditions for the KITTI and BDD-100K datasets.

Reviewer 2 Report
Comments and Suggestions for Authors
The study is generally successful. I recommend completing the following revisions.
The differences between the architectures shown in the figures (Figures 2-4) should be emphasized more clearly in the written explanation. Which layer was changed and why that layer was chosen should be explained in a clearly structured manner.
In the comparisons in Table 7, were all compared models tested on the same dataset and under the same conditions? This is unclear.
It is unclear whether this model was optimized only for the vehicle class. It is only included in the table but not explained in detail in the text. The "BaseSE-ECA (Vehicle)" Model Description is Insufficient.
In-Depth Analysis of BDD-100K Results is Missing. Performance degradation was observed, but no significant degradation was observed for which classes or sample outputs were provided.
Models such as YOLOv11n and YOLOv12n are new, but they are not included in the literature or detailed.
Ideas for future work exist, but they are very general ("spatiotemporal modeling"). More specific recommendations should be made.
Author Response
Comment 1: The differences between the architecture shown in the figures (Figures 2-4) should be emphasized more clearly in the written explanation. Which layer was changed and why that layer was chosen should be explained in a structured manner.
Response 1:
Thank you for highlighting the need for a more precise explanation of the architectural differences. In response, we have revised Section 3.2 of the manuscript to explicitly describe the changes made to each model and clarify the rationale behind the selected modifications.
- Figure 2 (BaseECA): We replaced the backbone's fourth and final C3 module with an ECA module. This layer appears just before the SPPF block, where high-level semantic features are aggregated. We selected this location to introduce lightweight channel attention at a critical feature refinement stage that influences the neck and detection head.
- Figure 3 (BaseECAx2): We integrated two ECA modules. The first replaces the backbone's C3 module at layer nine to enhance mid-level semantic representations early in the network. After the final C3 block in the detection head, the second is added to improve adaptive attention across the three detection scales (80×80, 40×40, and 20×20). This dual placement aims to improve multi-scale feature refinement with minimal added complexity.
- Figure 4 (BaseSE-ECA): We replaced the final C3 module in the backbone with an SE module to better capture channel interdependencies. The ECA module remains in the detection head to support localized attention during final prediction. This hybrid design combines the expressive strength of SE with the efficiency of ECA to balance detection performance and computational cost.
Bottom of Form
Comment 2:
In the comparisons in Table 7, were all the models compared tested on the same dataset and under the same conditions? This is unclear.
Response 2:
We thank the reviewer for highlighting this ambiguity. We confirm that all models presented in Table 7 (Now Table 8 in the revised version) were evaluated on the same dataset (KITTI). To ensure fair internal comparison, our proposed models were tested under identical experimental settings, including training epochs, input resolution, learning rate, optimizer, and data augmentation strategies.
For the external models listed in Table 8, results were obtained from their original published papers, which also used the KITTI dataset. However, as is common in benchmarking, the exact implementation settings (e.g., hardware, augmentation policies) may vary slightly between studies. To ensure transparency, we have clarified this point explicitly in the caption of Table 7 in the revised manuscript.
Comment 3:
It is unclear whether this model was optimized only for the vehicle class. It is only included in the table but not explained in detail in the text. The "BaseSE-ECA (Vehicle)" Model Description is Insufficient.
Response 3:
Thank you for pointing out the need for clarification regarding the BaseSE-ECA (Vehicle) model. We have revised Section 3.2 to describe this variant explicitly. In addition to the BaseSE-ECA model trained on all KITTI object classes, we developed a specialized version trained exclusively on the vehicle class, referred to as BaseSE-ECA (Vehicles). This version evaluates the model’s capability when focused on a single-class detection task.
The BaseSE-ECA (Vehicles) model leverages the same architecture as BaseSE-ECA but was optimized and fine-tuned using only the vehicle annotations from the KITTI dataset. The motivation was to assess whether class-specific training could enhance detection accuracy and efficiency for applications like intelligent transportation and surveillance. Our results show that the vehicle-specific variant achieved higher precision and recall for the vehicle class compared to the multi-class version, confirming the effectiveness of attention-based specialization.
This explanation has been added to Section 3.2 (BaseSE-ECA subsection) to provide a clearer description of the model’s purpose and performance context.
Comment 4:
In-Depth Analysis of BDD-100K Results is Missing. Performance degradation was observed, but no significant degradation was observed for which classes or sample outputs were provided.
Response 4:
We thank the reviewer for pointing this out. Upon further investigation, we found that the original BDD-100K results reported in Table 6; therefore, we updated it with corrected results obtained from the official test set.
To further address the reviewer’s comment, we added a new table (Table 7) presenting class-wise values measured by mAP@50 across all models, providing a more granular performance analysis under the complex and diverse conditions found in BDD-100K.
In the revised Section 5.1, we include a structured interpretation of these results. This class-wise analysis offers a clearer understanding of model limitations and helps guide future improvements.
Comment 5:
Models such as YOLOv11n and YOLOv12n are new but not included in the literature, or are not detailed.
Response 5:
Thank you for this critical observation. In response, we have revised the Related Work section to include YOLOv11 and YOLOv12 as recent advancements in the YOLO family. We summarized their architectural innovations, such as dynamic convolution aggregation and FlashAttention, and explained how they extend the design goals of previous YOLO versions.
Comment 6:
Ideas for future work exist, but they are very general ("spatiotemporal modeling"). More specific recommendations should be made.
Response 6:
We thank the reviewer for this insightful comment. In response, we have added a dedicated “Future Work” subsection in the revised manuscript to provide more specific and concrete directions. This new subsection outlines plans to enhance model robustness in challenging scenarios like those in the BDD-100K dataset. Specifically, we propose integrating temporal modeling techniques (e.g., ConvLSTM-based feature fusion and multi-frame transformer attention) to improve detection under motion and occlusion. We also suggest exploring domain adaptation methods such as style transfer and test-time training to simulate adverse weather and lighting conditions. Additionally, we plan to design controlled experiments to test the model’s resilience under sensor noise and environmental variability and validate these improvements using real-world datasets such as Cityscapes and nuScenes.

Reviewer 3 Report
Comments and Suggestions for Authors
- The paper specifies parameters such as learning rate, batch size, and augmentation techniques, but lacks an explanation of how these settings were selected based on specific criteria or prior experiments.
- The performance degradation on BDD-100K is simply attributed to the complex environment, but a more detailed analysis of the causes of the poor recall or the decrease in mAP for specific classes should be provided.
- If possible, it would be beneficial to evaluate the model's generalizability using additional datasets such as Cityscapes or COCO, in addition to KITTI and BDD-100K, for autonomous driving or general object detection tasks.
Author Response
Comment 1:
The paper specifies parameters such as learning rate, batch size, and augmentation techniques, but lacks an explanation of how these settings were selected based on specific criteria or prior experiments.
Response 1:
We appreciate the reviewer’s observation. In our study, we adopted the default training parameters provided by the official YOLOv5s implementation, including a learning rate of 0.01, batch size of 8, and standard data augmentation techniques. The original developers optimized these parameters for a wide range of object detection tasks and are commonly used as a reliable starting point in the literature.
To ensure these settings were also suitable for our models and datasets, we conducted preliminary experiments using alternative configurations, such as varying the learning rate, increasing the batch size, and modifying augmentation probabilities. However, these variations did not yield better performance or training stability. As a result, we retained the default configuration, which consistently produced the best trade-off between convergence speed, robustness, and accuracy.
We have updated Section 4.2 of the manuscript to include this explanation.
Comment 2:
The performance degradation on BDD-100K is attributed to the complex environment. Still, a more detailed analysis of the causes of the poor recall or the decrease in mAP for specific classes should be provided.
Response 2:
We thank the reviewer for this important suggestion. In response, we have added a new table (Table 7) that presents a class-wise breakdown of mAP@50 values across all models on the BDD-100K dataset. This addition allows us to analyze performance degradation at a finer level.
In the revised Section 5.1, we now provide a detailed interpretation of these class-wise results. Notably, the “Motor” and “Rider” categories exhibited the lowest mAP values (ranging between 21.1% and 25.1%, and 31.9% to 35.6%, respectively), which we attribute to the small object sizes, occlusions, and motion blur commonly found in these categories. In contrast, classes like “Car,” “Bus,” and “Truck” achieved higher detection accuracy, likely due to their larger object sizes and better visibility under various driving conditions.
This deeper analysis strengthens our understanding of where and why the models struggle on the BDD-100K dataset and supports our future work directions.
Comment 3:
It would be beneficial to evaluate the model's generalizability using additional datasets such as Cityscapes or COCO, in addition to KITTI and BDD-100K, for autonomous driving or general object detection tasks.
Response 3:
We fully agree that further evaluation on datasets such as Cityscapes or COCO would strengthen the generalizability analysis. Due to time and resource constraints, we limited our experiments to KITTI and BDD-100K, which already offer diverse real-world driving conditions. However, we have added this valuable recommendation to the Future Work section and plan to incorporate such datasets in our future extensions to assess model generalization across domains more rigorously.
